# Liquid marble-derived solid-liquid hybrid superparticles for $CO_2$ capture

Xia Rong[1], Rammile Ettelaie[2], Sergey V. Lishchuk [2,3], Huaigang Cheng[4], Ning Zhao[5], Fukui Xiao[5], Fangqin Cheng[4] & Hengquan Yang[1]

The design of effective $CO_2$ capture materials is an ongoing challenge. Here we report a concept to overcome current limitations associated with both liquid and solid $CO_2$ capture materials by exploiting a solid-liquid hybrid superparticle (SLHSP). The fabrication of SLHSP involves assembly of hydrophobic silica nanoparticles on the liquid marble surface, and co-assembly of hydrophilic silica nanoparticles and tetraethylenepentamine within the interior of the liquid marble. The strong interfacial adsorption force and the strong interactions between amine and silica are identified to be key elements for high robustness. The developed SLHSPs exhibit excellent $CO_2$ sorption capacity, high sorption rate, long-term stability and reduced amine loss in industrially preferred fixed bed setups. The outstanding performances are attributed to the unique structure which hierarchically organizes the liquid and solid at microscales.

[1] School of Chemistry and Chemical Engineering, Shanxi University, 030006 Taiyuan, China. [2] Food Colloids Group, School of Food Science and Nutrition, University of Leeds, Leeds LS2 9JT, UK. [3] Materials and Engineering Research Institute, Sheffield Hallam University, Sheffield S1 1WB, UK. [4] Institute of Resources and Environment Engineering, Shanxi University, 030006 Taiyuan, China. [5] State Key Laboratory of Coal Conversion, Institute of Coal Chemistry, Chinese Academy of Sciences, 030001 Taiyuan, China. Correspondence and requests for materials should be addressed to H.Y. (email: hqyang@sxu.edu.cn)

Carbon capture and storage (CCS) is increasingly recognized as an essential countermeasure against the current environmental and climatic challenges[1]. The discovery of efficient $CO_2$ capture materials is an ongoing major task to tackle this challenge[2,3]. As a benchmark carbon capture technology, amine scrubbing has a fast sorption rate, high sorption capacity and excellent selectivity[4]. This technology, nevertheless, suffers from amine evaporative loss, equipment corrosion, and high operational cost due to the inherent shortcomings of liquids[5]. As an alternative technology, solid sorbents have been extensively investigated because of their inbuilt advantages such as non-corrosion and less environmental impacts[6–9]. Among various reported solid sorbents, amine-supported solid materials are very attractive for practical applications because of the excellent capability to capture low-concentration $CO_2$ (<15%). This class of sorbents is prepared by either impregnating porous supports with tetraethylenepentamine (TEPA) [or polyethyleneimine (PEI)][10,11], or grafting the amines on supports[12,13]. Despite impressive advances, the amine-impregnated sorbents suffer from amine loss during multiple cycles[11,14], and the amine-grafted sorbents usually require multiple steps of molecular modification. Another interesting class of sorbents, involving encapsulation of liquid sorbent droplets with solid shell, has recently been proposed[15–19]. For example, Cooper et al. and our group reported a "dry water" adsorbent, in which $K_2CO_3$-containing droplets were coated by a layer of hydrophobic particles[15,17]. Vericella et al.[20] prepared a sorbent by encapsulation of an aqueous carbonate core with a polymer shell. Despite considerable achievements, these methods are only suitable for aqueous based systems, but not appropriate for neat amines because of the extremely high viscosity which leads to very low sorption capacity and rate. This crucial weakness hinders their practical applications.

Another important concern is the size and morphology of sorbents because of the great relevance to any practical utilization[21–23]. Generally, spherical particles with sizes ranging from dozens of microns to several millimeters are preferred because such large particles can directly be installed in fixed- or fluidized-bed reactors, working with low pressure drop and reduced attrition[24]. To meet these technical requirements, the solid sorbents with small sizes are shaped to the large-sized objects with uniform morphology through compression[25–27] or spray-drying with assistance of binders[28–30]. However, the addition of binders often leads to unpredictable structures and significant decrease in sorption capacity. For the micron-to-millimeter sized silica, the highest $CO_2$ sorption capacity ever recorded, was only 4.3 mmol g$^{-1}$ (using 99% $CO_2$ for adsorption test)[31]. Even worse, the amine loss during repeated cycles is not negligible. This is a common problem associated with all the reported amine-impregnated sorbents[28,32]. In this context, development of large-sized spherical sorbents with high sorption capacity, fast sorption rate, low amine loss and high mechanical strength remains an urgent yet unfulfilled challenge.

Here, aiming to merge the respective merits of both liquid and solid materials, we detail a concept on a solid–liquid hybrid superparticle (SLHSP) sorbent (Fig. 1), made of a hydrophobic porous shell and a TEPA-mesoporous silica composite core where liquid and solid are hierarchically organized at a microscale level. Distinctive features of such a structure are multifold: the highly viscous TEPA is compartmentalized into tiny droplets, achieving a high surface-to-volume ratio; the accessibility of TEPA is further enhanced due to the presence of "pores" inside the liquid; the mechanical strength is dramatically reinforced through solid–liquid interactions; the amine loss and equipment corrosion are significantly suppressed due to the presence of a hydrophobic shell. Our method is demonstrated to enable the reliable control over the interior structure, size and mechanical strength of

SLHSP. The developed sorbent works well in industrially preferred fixed-bed reactors with a high working capacity, long-term stability and outstanding ability to retard the amine loss. The fundamental principles underpinning the formation of SLHSPs and the insights into their sorption behaviors are experimentally and theoretically investigated.

## Results

**SLHSP design and preparation.** To shape a solid–liquid mixture into micron-to-millimeter sized microspheres, we harnessed a so-called liquid marble phenomenon as it offers the possibility to compartmentalize liquid droplets within a layer of hydrophobic nanoparticles[33–35]. We chose mesoporous silica particles (MSPs) with large size, high porosity and low density, modifying them with octyltrimethoxysilane (MSP-Os) to serve as hydrophobic nanoparticles. The reason for this special option is that large-sized particles possess high adsorption energies when placed on interfaces, according to the established equation[36],

$$\Delta E = \pi R_s^2 \gamma_{aw} (1 \pm r \cos \theta)^2 \qquad (1)$$

where $R_s$ is the radius of the particle, $\gamma_{aw}$ is the interfacial tension between air and water, and $r$ is the surface roughness factor. The great adsorption energy is beneficial to enhancing the robustness of liquid marbles. Another important consideration is that the low density can reduce the gravity effects that may cause the particle dislocation on the droplet surface. As shown by scanning electron microscopy (SEM) and transmission electron microscopy (TEM) images in Fig. 1a, MSPs have particle sizes from hundreds of nanometer to one micron, and their pore size is centered around 15–19 nm, which is broadly consistent with the BJH pore size (Supplementary Fig. 1). After modification the surface of MSP changed from hydrophilic to hydrophobic one, with the water contact angle increasing remarkably from 20º to 135º (Supplementary Table 1). [$N_2$ sorption isotherms, Fourier transform infrared spectroscopy (FT-IR), thermogravimetric analysis (TGA) of MSP-Os, Supplementary Fig. 1].

We first discuss the preparation of aqueous TEPA marbles (without incorporation of MSP). The preparation procedure mainly involves dropping of 30 wt% aqueous TEPA onto a bed made of MSP-Os with the assistance of a syringe pump, followed by rolling of the droplets. Due to the gas/liquid interfacial adsorption, the hydrophobic MSP-Os were picked up by the surface of droplets, hence forming a layer of hydrophobic nanoparticles. The resultant aqueous TEPA marbles is a free-flowing powder (Supplementary Fig. 2a and b). Even after standing for 15 days, they maintained their spherical morphology (Supplementary Fig. 2c), indicating high stability against structural rupture. In contrast, for the aqueous TEPA marbles prepared with another mesoporous silica MSNs (with pore size of 8 nm, modified with the same amount of octyl organosilane, Supplementary Fig. 3a), their surfaces were observed to be partially bare even as early as 7 min following their preparation (Supplementary Fig. 3b). Another distinction between the two kinds of liquid marbles is their fusion behavior. When a few liquid marbles prepared with MSNs were brought into contact with each other, the fusion of liquid marbles occurred (Supplementary Fig. 3c). In contrast, the liquid marbles prepared by MSP-Os remained fully separated, which is attributed to the fact that MSP-Os assembled on the surface provide a perfect shield (without any bare areas on the surface). These results verify that formation of such a shield on the surface is crucial for the realization of high-quality aqueous TEPA marbles.

SLHSPs were prepared through the same procedures as above (Fig. 1), except for addition of 23 wt% of MSP into the aqueous TEPA solution at the start of the procedure (the percent refers to

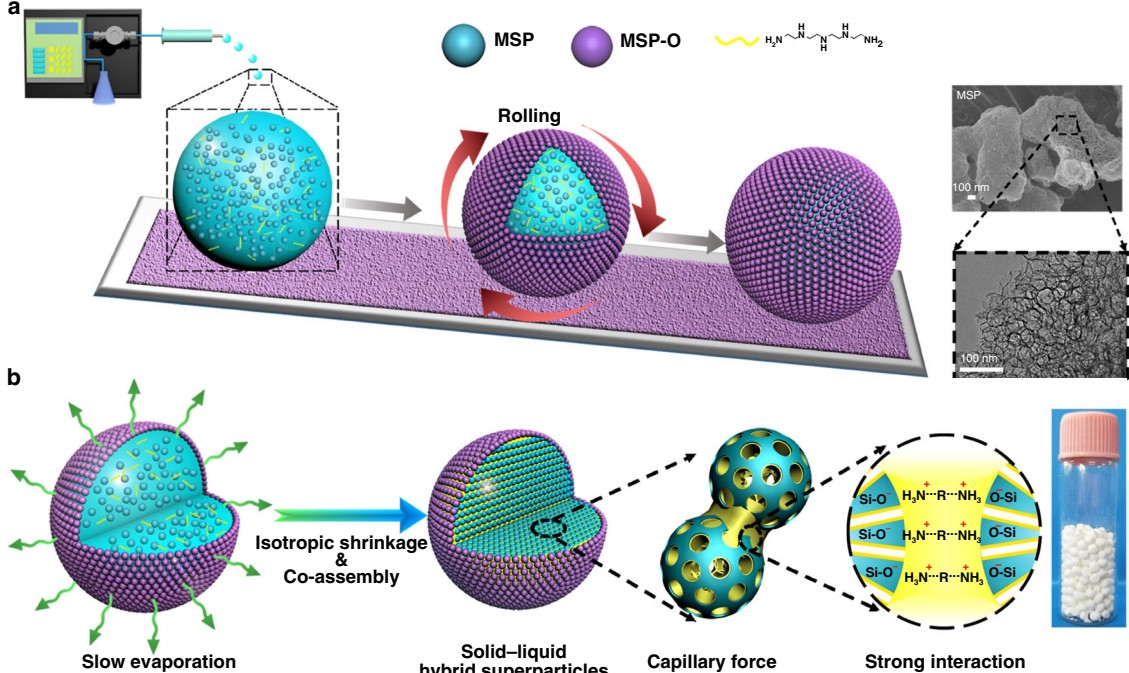

**Fig. 1** Schematic illustration of the preparation of SLHSPs. **a** Dispensing MSP-containing aqueous TEPA droplets using a syringe pump and rolling the droplets on a bed made of MSP-Os, SEM image of MSP (upper panel), TEM image of MSP (lower panel). **b** Water evaporation causing isotropic shrinkage of the marbles and co-assembly of MSP with TEPA, and appearance of SLHSPs (in a vial)

the mass fraction of MSP in the aqueous TEPA solution). Slow evaporation of water inside the marbles caused isotropic shrinkage of the marbles (Fig. 1b). Consequently, MSP-Os got tightened due to the strong interfacial adsorption, while the inner TEPA and MSPs were subjected to assembly due to the hydrogen-bonding interactions[37,38]. In contrast, evaporation of the suspension droplet (without MSP-Os on the surface) led to a doughnut-like shape due to the so-called coffee loop effect (Supplementary Fig. 4)[39,40]. This comparison demonstrates the rather important role of the presence of MSP-Os in controlling morphology. The underlying reason may be that the hydrophobic MSP-Os layer slowed and ensured a more uniform water evaporation rate everywhere on the surface of droplets, such that the mass transport rate within the interior could keep up with the shrinkage rate of droplets.

**Characterization**. As the SEM images in Fig. 2a show, the obtained SLHSPs are composed of discrete and uniform microspheres with an average size of 600 μm. MSP-Os are closely packed on the outer shell, and plenty of apertures between MSP-Os appear on the shell. The SEM image of the cross-section reveals existence of plenty of macropores stemming from the interstices between MSPs in spite of some agglomeration of primary particles. Obviously, without MSP (TEPA alone) it is impossible to obtain a fraction of void within the interior. Taking into account the mesopores of MSP per se, a hierarchical mesoporous–macroporous network was formed inside the interior of SLHSP, as confirmed by the SEM and TEM images of the calcined SLHSP (Supplementary Fig. 5). After calcination the hierarchical pores are far more evident. The locations of MSP-Os and MSPs were investigated through fluorescence dyeing experiments. When MSP-Os were labeled with a fluorescent reagent, isothiocyanate isomer, a fluorescent circle around the superparticle was clearly observed (Fig. 2b), indicating that MSP-Os were distributed solely on the surface of SLHSP. However, when MSPs were labeled with the fluorescent reagent, fluorescent

signals were found to be present throughout the interior of SLHSP, confirming that MSPs were positioned within the interior. As elemental mappings (Fig. 2c) show, Si and N elements are uniformly distributed on SLHSP, underscoring that the solid and liquid are well organized at a microscale level. Such a homogeneity should be attributed to the strong hydrogen-bonding interactions between TEPA molecules and the surface silanols of MSP[10]. Figure 2d displays the TGA results of SLHSP. From 120 to 300 °C, there is a major weight loss (ca. 80.3 wt%) caused by the evaporation or decomposition of TEPA, which is consistent with the amount added initially.

As seen from the FT-IR spectrum of SLHSP (Fig. 2e), N–H stretching vibration (3368, 3302 cm⁻¹), N–H bending vibration (1595 cm⁻¹), C–H stretching vibration (2931, 2881 cm⁻¹) and C–H bending vibration (1457, 1346 cm⁻¹) were all clearly found, indicating successful encapsulation of TEPA. Figure 2f shows the ¹³C NMR spectrum of liquid TEPA, and the solid state ¹³C CP-MAS NMR spectra of SLHSP and the TEPA grafted on MSP (the procedure for grafting TEPA onto MSP is described in Supplementary Method 2). The peaks for SLHSP are significantly broadened compared to the peaks of the liquid TEPA, but are still relatively narrower than those for the TEPA grafted MSP. This finding is a proof that the mobility of TEPA in SLHSP is somewhere in between that of the liquid TEPA and the grafted TEPA, reflecting an important feature of our hybrid material. Accordingly, it can be concluded that the TEPA in SLHSP has a liquid-like behavior to some extent.

Fluorescent reagents were used to visualize the permeability of SLHSP to external molecules. Rhodamine B, initially dissolved in a solution (a mixture of ethanol and water, 1:1 V/V), was observed to spontaneously enter SLHSP, as demonstrated by the time-dependent fluorescence microscopy (Fig. 2g). The fluorescent intensity within the interior area gradually increases as a result of the diffusion of more fluorescent molecules into its interior, and then levels off after about 1.2 min. This excellent accessibility to external molecules is due to the unique

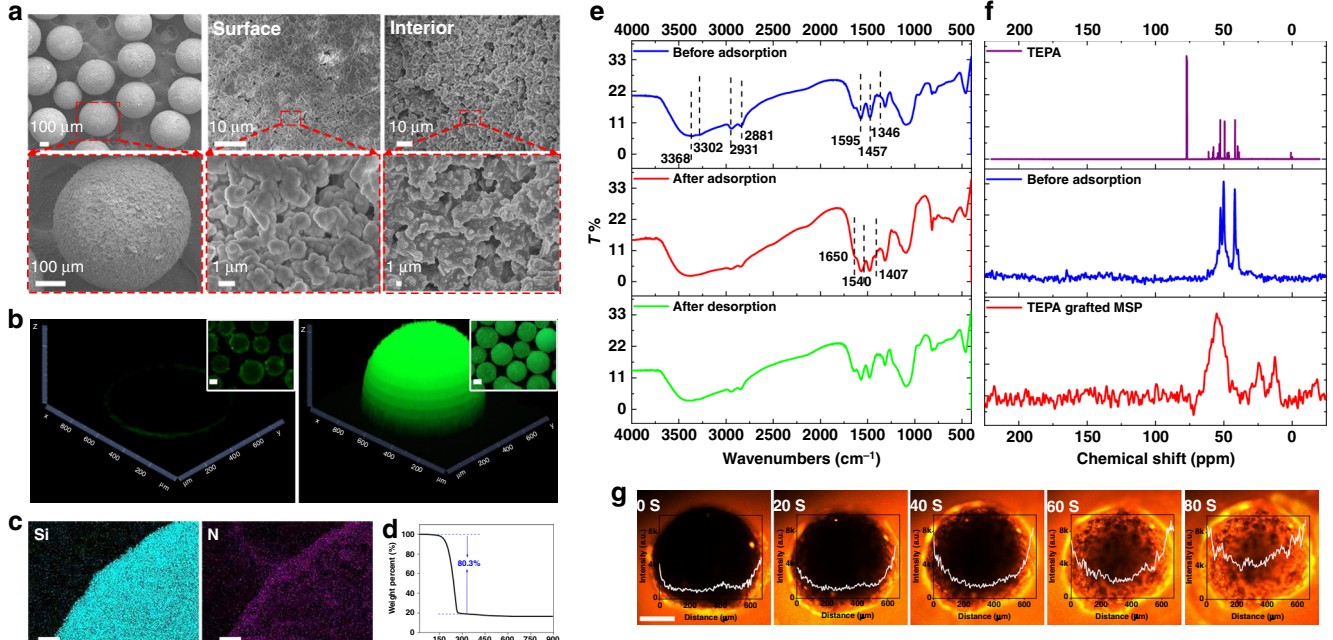

**Fig. 2** Characterization of SLHSPs. **a** SEM images showing the morphology, surface and interior structure of a SLHSP cut open deliberately. **b** Fluorescence confocal microscopy image of SLHSP with MSP-Os (left) and the inner MSPs (right) dyed by fluorescein isothiocyanate isomer, respectively. **c** Elemental mappings by SEM, scale bar = 1 μm. **d** TGA profiles of SLHSP. **e** FT-IR spectra of SLHSP before/after $CO_2$ sorption and after desorption. **f** $^{13}$C NMR of liquid TEPA and $^{13}$C CP-MAS NMR spectra of SLHSP and the TEPA grafted MSP. **g** Time-dependent fluorescence intensity for the transport of Rhodamine B into a SLHSP, scale bar = 200 μm

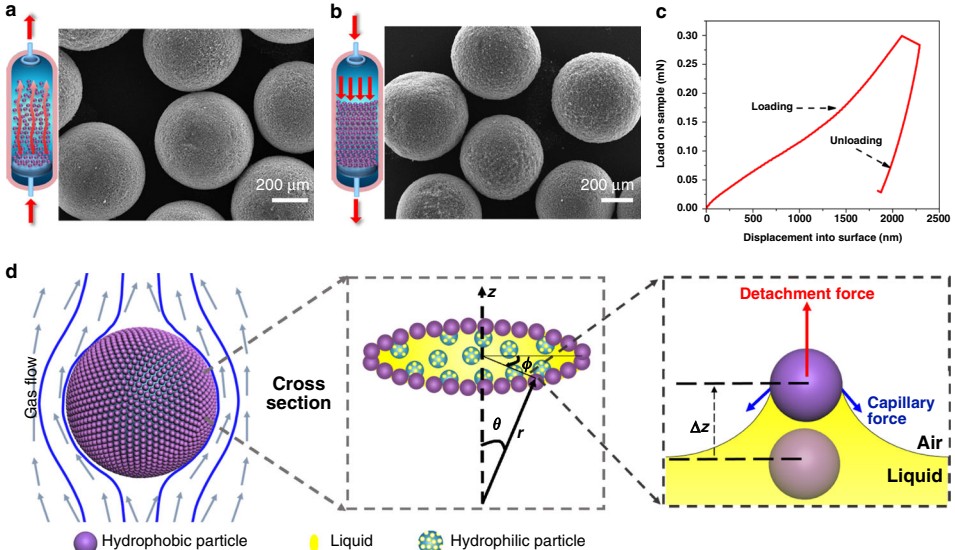

**Fig. 3** SLHSP mechanical strength test and theoretical analysis. **a** SEM image after the test in a fluidized-bed reactor. **b** SEM image after the test in a fixed-bed reactor. **c** Load-displacement diagram of a single SLHSP. **d** Force analysis for MSP-O at the surface in a flow gas

architecture of SLHSP, which integrates macropores with mesopores in the interior of superparticles.

**Mechanical stability**. SLHSPs were subjected to be fluidized in $N_2$ at a superficial velocity ca. 0.3 m s$^{-1}$ (Fig. 3a). After 3 h, SLHSPs retained their original morphology and structural integrity without any detectable fractures, as evident from the SEM image. Meanwhile, SLHSPs were filled in a fixed-bed reactor and treated with $N_2$ at a flow rate of 25 ml min$^{-1}$ and a pressure of 2 MPa for 3 h (Fig. 3b). After treatment, SLHSPs also maintained their

structural integrity. These results confirm that our so-designed SLHSPs are amenable to utilization in fluidized- or fixed-bed reactors, as anticipated.

To clarify the underpinning principle that governs the robustness of SLHSP, we determined the mechanical properties of SLHSP at a nanoscale level with the nanoindentation technique[41]. The indentation load-displacement behavior of SLHSP tested with a Berkovich indenter has been carefully documented. As Fig. 3c displays, SLHSP exhibits a hysteresis loop in the indentation load-displacement curve, as might be expected

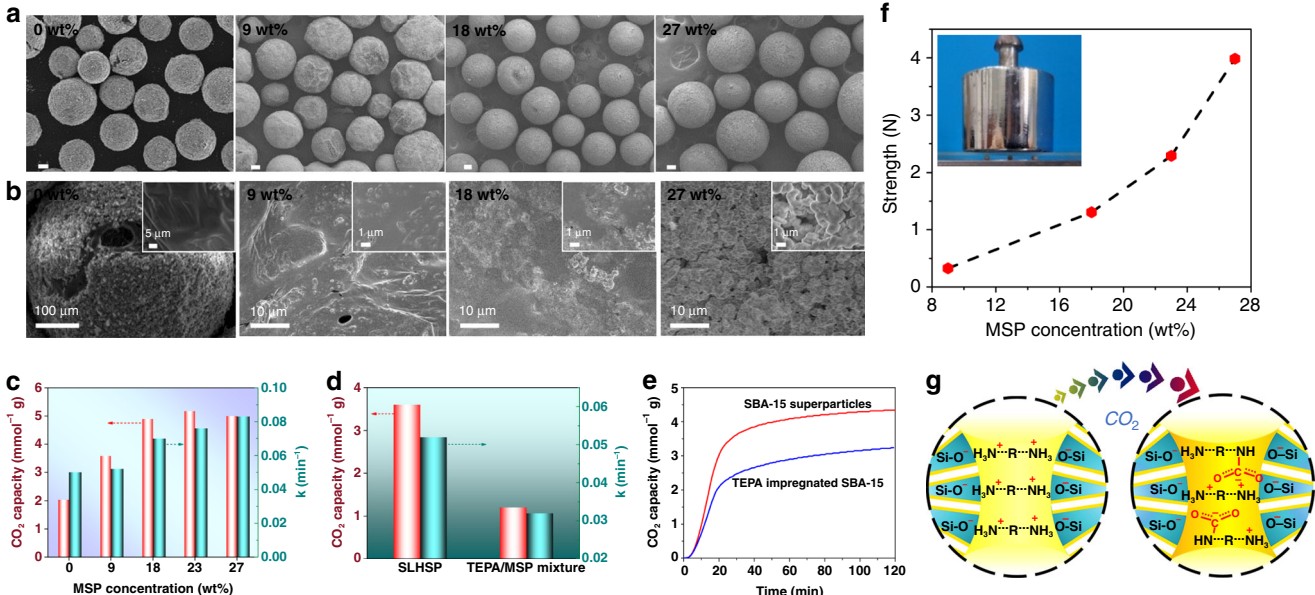

**Fig. 4** Structure tuning and performances of SLHSPs with different amounts of MSP. **a** SEM images of the morphology of the liquid TEPA marble and SLHSPs with different amounts of MSP (9, 18 and 27 wt%), scale bar = 100 μm. **b** SEM images of the cross-section of the liquid TEPA marble and SLHSPs with different amounts of MSP. **c** $CO_2$ sorption capacity and rate for the SLHSPs containing different amounts of MSP. **d** $CO_2$ sorption capacity and rate of SLHSPs containing 9 wt% of MSP and the TEPA/MSP mixture with the same MSP. **e** Time-dependent $CO_2$ uptake profiles of the superparticles made of SBA-15 and TEPA-impregnated SBA-15. **f** Mechanical strength of a single SLHSP as function of the MSP amount after $CO_2$ sorption. **g** A model showing the change of interactions between silanol groups and amines after $CO_2$ sorption

a portion is recovered on unloading. This finding indicates that SLHSP behaves elastically to some extent[41]. The interfacial adsorption force and capillary force always act on MSP-O[42], such that the superparticle tries to preserve a spherical shape, thereby being elastic. At the same time, within the interior, MSP acts as pillar to support the network, while TEPA serves as connector or glue to hold the particles together because of the high affinity of TEPA to the silica surface, as reported in the system of a mixture of particles and polymers[43]. Owing to these strong interactions between liquid and solid, in spite of some deformation, the structural integrity was still maintained when experiencing external compression.

To quantify the magnitude of the limiting flow rates which SLHSPs can withstand, we established the following physical model (Supplementary Note). We assume that the set of stabilizing nanoparticles, having dimensions much smaller than the hybrid superparticle, can be considered as a continuous thin spherical shell around the superparticle of radius $R$. The flow of the continuous phase around this superparticle exerts a hydro-dynamic force $f$ on its surface, causing development of stresses within the shell (Fig. 3d). The magnitude of the stress in this thin shell is calculated, taking into account that in the absence of any attractive forces between the nanoparticles, the shell cannot support extensional stresses. Expressed in spherical polar coordinates, the stress, $f_θ$, at angle $θ$ is shown to be

$$f_e = \frac{3\eta u}{2R} \sin\theta \qquad (2)$$

where $η$ denotes the viscosity of the continuous phase. The maximum stress occurs at the bottom of the sphere ($θ = 180°$), where one finds

$$\sigma_{\theta\theta}^{(S)} = \frac{3\eta u}{h} \qquad (3)$$

A nanoparticle placed on the surface at the bottom of a superparticle will consequently experience compressive forces exerted by it by its surrounding neighbors. When the particle is

slightly displaced from its equilibrium by a small distance $Δz$, a radial force develops.

$$F_{\text{hyd}} = 6\pi\eta u\Delta z \qquad (4)$$

Opposing this are the capillary forces pulling the particle back. For small displacements, such capillary forces have a linear dependence $F_{\text{cap}} = k_s\Delta z$, where the constant $k_s$ is sometimes known as the Hooke-de Gennes spring constant and was calculated to be

$$k_s = \frac{4\pi\gamma}{2\ln(2R/h) \pm 1} \qquad (5)$$

for spherical particles sitting on droplets[44]. Here $γ$ is the interfacial tension between the dispersed and dispersion phases, with the ± sign referring to pinned and unpinned contact lines. From the above equations, the limiting flow $u^*$ is determined when $k_s < 6\pi\eta u$, i.e.:

$$u^* = \frac{\gamma}{3\eta\left(\ln(2R/h) \pm \frac{1}{2}\right)} \qquad (6)$$

Using this result, the limiting flow rate at which the nanoparticles are displaced is 100.9 m s$^{-1}$, for gas with low viscosity of 0.018 mPa s, $R/h = 600$ and $γ = 41.1$ mN m$^{-1}$. This remains well above typical values encountered in commercial fluidized beds (1 m s$^{-1}$)[45].

**Structural tuning and structure–performance relationship.** To identify the key parameters for the structural tuning and to clarify the structure–performance relationship, we prepared two sets of SLHSPs with different amounts of MSP or with different sizes. By varying the initial fraction of MSP in the suspension, 9–27 wt% MSP could be readily incorporated into SLHSPs. As shown in Fig. 4a, the morphology and size of the resultant SLHSPs strongly depend on the MSP amount. Without incorporation of MSP, there are cracks on the surface of TEPA marbles, indicating that the neat liquid marble is not very stable during the SEM obser-vation. In contrast, SLHSPs with 9 wt% MSP exhibited a distinct

spherical morphology despite of some winkles on their surface resulting from the buckling during the drying process[38]. When the MSP amount was increased up to 18 and further to 27 wt%, a perfectly spherical morphology with smooth surface was achieved. Aside from this change, the superparticle size was also altered. The size increased from 200 to 300, and 600 μm when MSP changed from 9 to 18, and 27 wt% (Supplementary Fig. 6a–e). This size change could be explained in terms of the increasing viscosity. The viscosity of a dispersion with a low solid volume fraction $\phi$ is given by the well-known Einstein equation [46]:

$$\eta_r = (1 + 2.5\phi) \qquad (7)$$

where $\eta_r$ is the relative viscosity defined as a ratio of the suspension viscosity to the viscosity of the suspending medium. Accordingly, the higher the volume fraction of MSP is, the greater the viscosity of the suspension is. The drop formation is related to the balance between two forces: the disruptive force attempting to break the droplets in an applied shear flow, and the resistive forces originating from the interfacial tension and the viscosity of dispersed phase (in Methods). The increase in viscosity provides a larger resistance to breakage, thus leading to larger droplets. These results demonstrate that MSPs play a particular role in controlling the size as well as the morphology. Moreover, the MSP amount profoundly impacts the interior structure of SLHSPs, as shown by the SEM images of the cross-sections (Fig. 4b). For the TEPA marbles (without MSP), one can see the interior is "empty". When tuning the MSP amount upwards, the irregular pores with micron-leveled sizes were observed. These porous structures were more pronounced especially for SLHSPs with 23 and 27 wt% MSP. Such pores are void spacing formed due to the loose aggregation of MSP within SLHSP. It is self-evident that the choice of a proper proportion of liquid to particle is crucial to obtaining the hierarchical interior structures.

It is interesting to compare the $CO_2$ sorption performances of the SLHSPs with different internal structures (using a simulated flue gas of 15 vol % $CO_2$ in $N_2$). As presented in Fig. 4c, the $CO_2$ sorption capacity of the TEPA marble is only 2.0 mmol g$^{-1}$. Once MSP (9 wt%) is introduced, the sorption capacity is sharply improved, up to 3.6 mmol g$^{-1}$. Further tuning of the MSP amount upward to 18 and 23 wt%, the $CO_2$ sorption capacity is seen to continuously increase reaching as high as 4.9 and 5.2 mmol g$^{-1}$. However, when 27 wt% MSP was included, the sorption capacity decreased slightly (5 mmol g$^{-1}$). More interestingly, these SLHSPs exhibited different sorption kinetics (Supplementary Fig. 6f). To qualify these differences, we fitted the sorption profiles with pseudo-first-order kinetic equation (see Method)[30,47]. The calculated sorption kinetic $k_1$ increases from 0.050 to 0.052, 0.070, 0.076, 0.083 min$^{-1}$ upon increasing the amount of MSP correspondingly from 0 to 9, 18, 23, 27 wt% (Fig. 4c). These results clearly show that the incorporation of more MSPs can effectively improve both sorption capacity and sorption rate. The MSPs within the interior of SLHSPs not only create large pores but also benefit dispersion of the viscous TEPA, thereby significantly facilitating the accessibility. Besides the amount, the pore structures of the chosen nanoparticles are also important. For example, if MSP was replaced by a typical mesoporous silica SBA-15 with smaller pores (7 nm, Supplementary Fig. 7a–c), the sorption capacity was only 4.3 mmol g$^{-1}$ (Fig. 4e), which was much lower than that of SLHSP with 23 wt% MSP. This finding further confirms that the interior structure of superparticles dictates the sorption performances. The presence of hydrophobic nanoparticles on the surface of superparticle is also crucial. For example, in the absence of MSP-Os, the sorption capacity and rate of the TEPA/MSP mixture (with 9 wt% MSP) are both much lower than that of SLHSP with the same amount of MSP (1.2 vs 3.6 mmol g$^{-1}$; 0.032 vs 0.052 min$^{-1}$, Fig. 4d). The

comparison of other pairs of samples further supports this conclusion (Supplementary Fig. 6g and h). Such a striking contrast is also manifested in the case of SBA-15 (Fig. 4e, 3.2 vs 4.3 mmol g$^{-1}$; 0.053 vs 0.063 min$^{-1}$). The presence of the hydrophobic layer can compartmentalize the gel-like TEPA/MSP (or SBA-15) mixture into numerous microspheres (Supplementary Figs. 6i and 7d). The enlarged gas/solid surface area and thereof shortened $CO_2$ transport distances account for the enhanced sorption performances. Moreover, the amine efficiency (the $CO_2$/N ratio) of the SLHSP with 23 wt% MSP is estimated to be 0.25, which is lower than the theoretical amine efficiency (0.5 $CO_2$ per mol N under dry conditions). Although the amine efficiency is lower than that of the reported sorbents prepared by grafting amine, it is slightly higher than those of nanosized amine-impregnated sorbents (0.18) and our amine-impregnated MSP (0.23)[48]. The $CO_2$/$N_2$ selectivity is estimated to be up to 187 from the TG-MS measurement (Supplementary Fig. 8). The high selectivity originates from the strong $CO_2$–amine chemical interactions.

The superparticle size is also tunable by changing the feeding rate of the suspension. As shown in Fig. 5a–d, the average diameter of SLHSP with 23 wt% MSP could be regulated from 150 to 500 μm upon increasing the flow rate from 0.05 to 0.3 ml min$^{-1}$. More interestingly, superparticles as large as 2 mm were achieved through vibrating the needle of syringe pump. It is worthwhile to emphasize that such a wide size range is essential in practical applications though remains very challenging to achieve using other currently existing methods[49]. The SLHSPs also displayed different sorption kinetics and capacity (Fig. 5e). The sorption kinetics ($k_1$) increased with decreasing size (Fig. 5f) because the decrease in the sorbent size results in not only an increase in the total gas–amine interface area but also a reduction in the diffusion distance. Conversely, the sorption capacity increases with increasing the superparticle size. This is explained by the fact that when increasing the superparticle size the fraction of MSP-Os on the superparticle decreases (MSP-O per se cannot adsorb $CO_2$, Supplementary Fig. 9).

Another intriguing discovery is the self-stiffening behavior of SLHSP after sorption of $CO_2$. As shown in Fig. 4f, with increasing the amount of MSP from 9 to 18, 23, 27 wt%, the mechanical strength for a single particle reaches 0.3, 1.3, 2.3 and 3.4 N after completing $CO_2$ sorption. The mechanical strength of the latter two is even comparable to the conventional solid particles[50], which can be explained by the fact that there exist strong electrostatic interactions in the interior network (Fig. 4g). As revealed by the FT-IR in Fig. 2e, after $CO_2$ sorption, new absorbance bands at 1650, 1540, and 1407 cm$^{-1}$ appear, which can be assigned to N–H bending vibration in R–$NH_3^+$, C=O stretching vibration and NCOO$^-$ skeletal vibration, respectively[11]. The $^{13}$C CP-MAS NMR spectrum (Supplementary Fig. 10) further confirms the formation of carbamate as a peak at 164.6 ppm appears. The strong electrostatic interactions between R–$NH_3^+$ and –NCOO$^-$, R$NH_3^+$ and surface Si–O$^-$ come into play[51], thus leading to a significantly enhanced rigidity. Interestingly, it was found that a single SLHSP could resist a mechanical compression of 2 N after few minutes exposure to the simulated flue gas. Such a self-stiffening behavior can be very important in practical applications.

**Multiple adsorption–desorption cycles**. To examine the long-term stability of SLHSP, multiple adsorption–desorption cycles were conducted. Being micro-sized spheres, SLHSPs could be easily packed in a fixed-bed reactor, without need for a shaping process often necessary for nanometer-sized sorbents, as illustrated in Fig. 6a. After being activated in $N_2$ at 100 °C for 60 min,

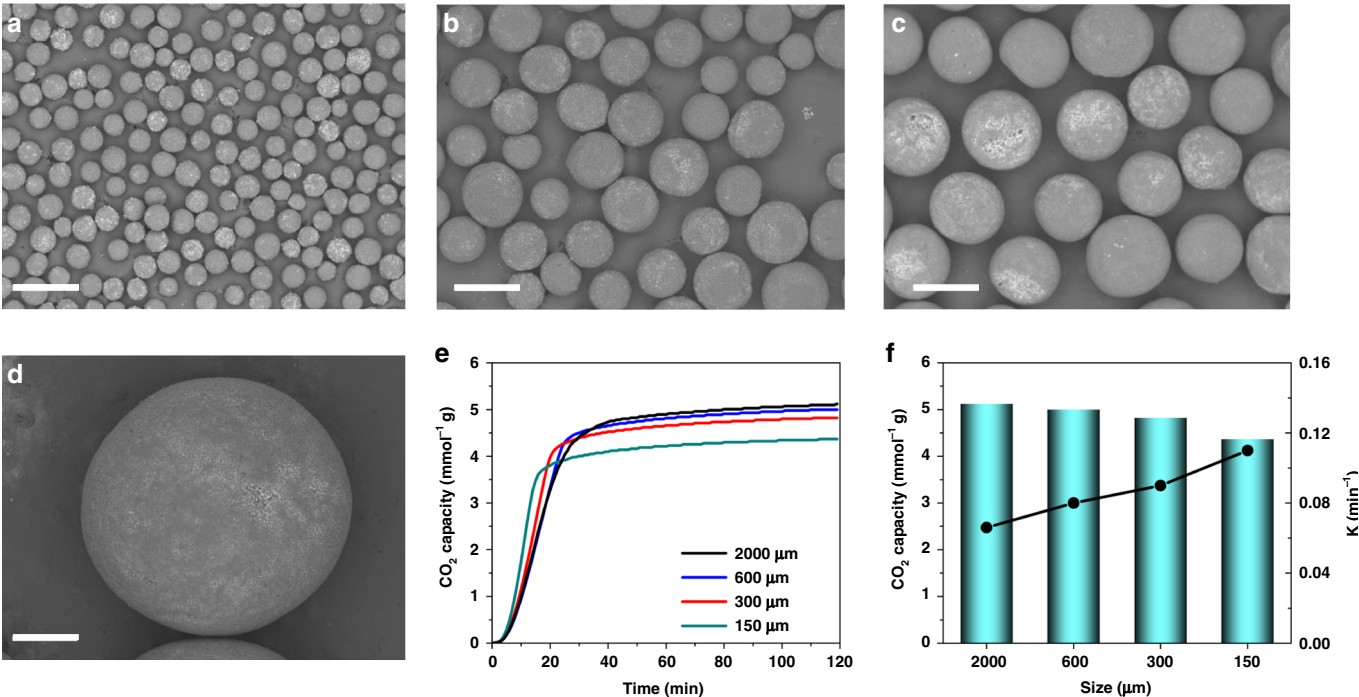

**Fig. 5** SEM images of SLHSPs (23 wt% MSP) with different sizes and their $CO_2$ sorption performances. **a** 150 µm, **b** 300 µm, **c** 500 µm, **d** 2000 µm, Scale bar = 500 µm. **e** Time-dependent uptake profiles of SLHSPs with different sizes. **f** Sorption kinetics ($k_1$) and capacity ($Q_e$) of SLHSPs with different sizes

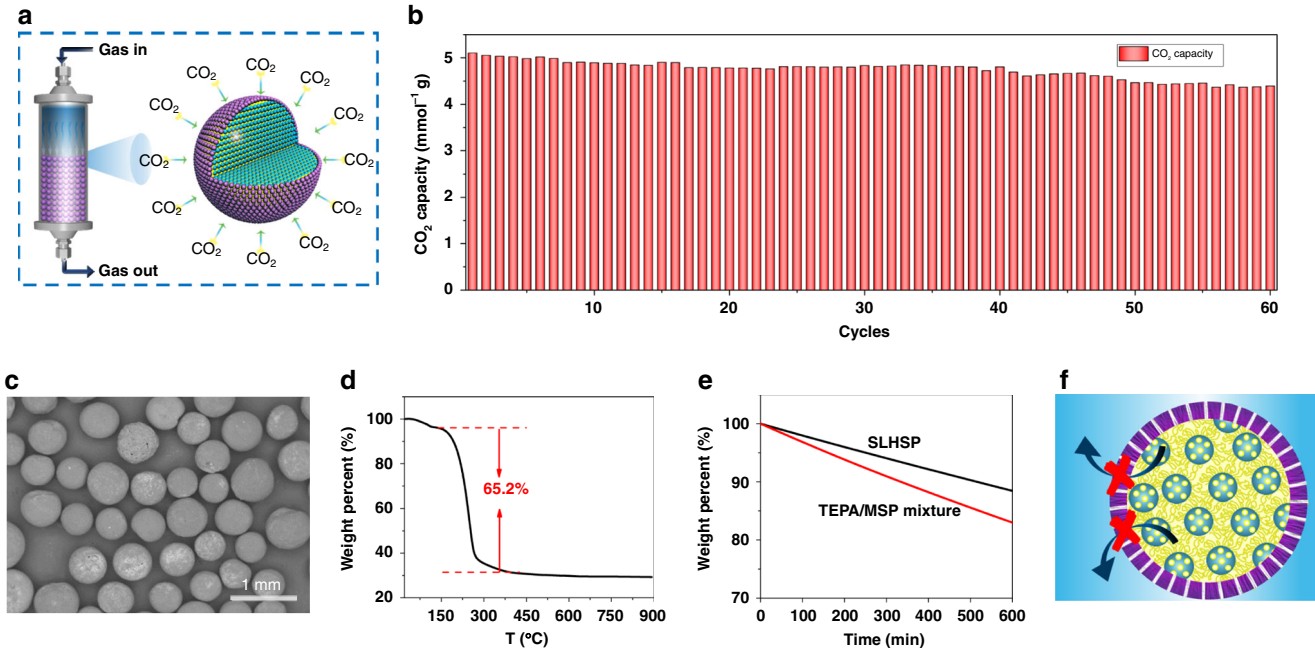

**Fig. 6** Multiple adsorption–desorption cycle test of SLHSP. **a** Schematic illustration of the $CO_2$ sorption in a fixed-bed reactor. **b** Multiple cycles of $CO_2$ adsorption–desorption over SLHSP. **c** SEM images of the spent SLHSP after 60 cycles. **d** TGA profile of the spent SLHSP after 60 cycles. **e** Comparison of the weight loss between SLHSP and TEPA-impregnated MSP at 100 °C in flowing $N_2$. **f** Carton showing that the hydrophobic outer layer of SLHSP prevents the penetration of TEPA molecules through the shell

the simulated flue gas at a flow rate of 25 ml min$^{-1}$ was allowed to pass through the bed for $CO_2$ capture (at 75 °C). The $CO_2$ sorption capacity of the fresh SLHSP was determined to be as high as 5.1 mmol g$^{-1}$ (Fig. 6b), which is close to the value determined by the TGA method (5.2 mmol g$^{-1}$). Such a sorption capacity is considerably higher than that of the previous reported

amine-imprecated mesoprous micron-sized spheres[28,29] and nanospheres, as summarized in Supplementary Table 3. The enhanced sorption capacity is ascribed to the unique structure that hierarchical organization of liquid and solid at a microscale level within our superparticles. The absorbed $CO_2$ could be nearly completely desorbed after being treated with $N_2$ at 100 °C. In the

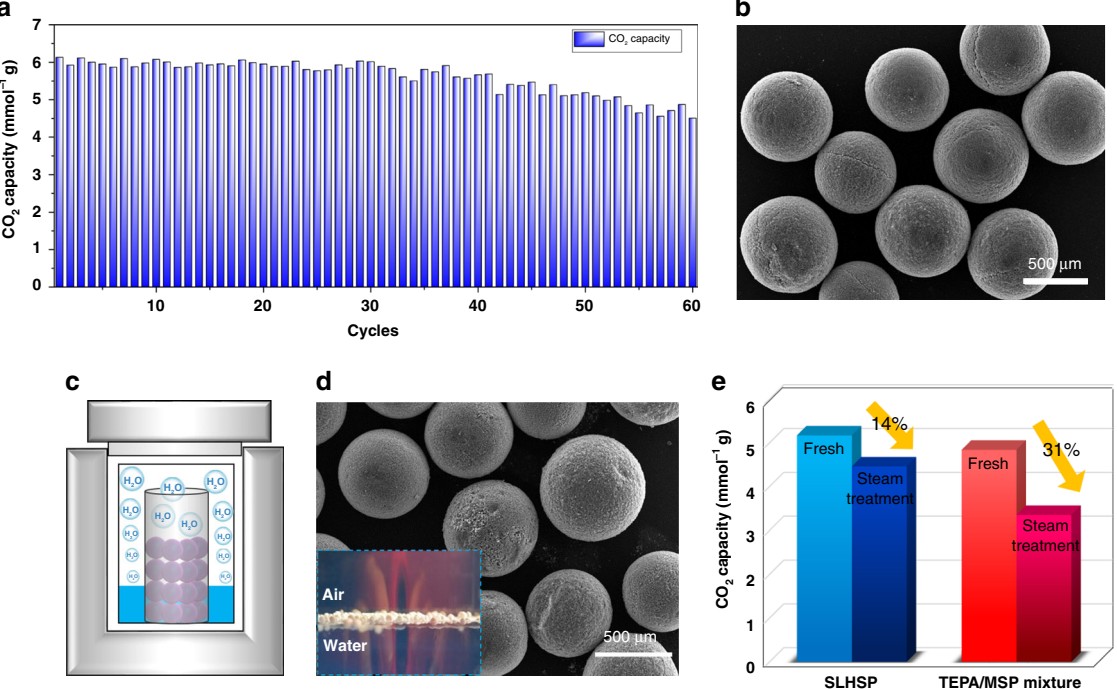

**Fig. 7** Multiple adsorption–desorption cycle test of SLHSP in the presence of moisture. **a** Cyclic sorption capacity of SLHSP in a fixed-bed reactor in pre-humidified 15% $CO_2$. **b** SEM of the spent SLHSP after 60 cycles. **c** Schematic apparatus for the steam treatment of SLHSP and TPEA-impregnated MSP. **d** SEM of the SLHSP after the steam treatment with a mixture of water steam at 106 °C for 12 h. **e** Sorption capacity loss of SLHSP and TEPA-impregnated MSP after steam treatment

second cycle of sorption, the regenerated SLHSPs were found to be able to offer a sorption capacity of 5.06 mmol g⁻¹, without any apparent decrease in comparison to the fresh ones. The adsorption–desorption cycles smoothly proceeded without any appreciable decline in sorption capacity in the followed cycles. After 60 cycles, a sorption capacity of 4.4 mmol g⁻¹ was achieved, which is still 86% of the initial capacity. Furthermore, the SLHSP recovered from the 60th cycles remained intact without any obvious deformation and rupture, as shown in Fig. 6c. The loss of sorption capacity is mainly due to the TEPA loss, as the TGA results revealed a 15 wt% amine loss after 60 cycles relative to the fresh SLHSP (Fig. 6d). To clarify the structural and compositional changes of SLHSP over multiple cycles, we characterized SLHSPs after 60 cycles with SEM, TEM and N₂ physical sorption analysis (Supplementary Fig. 11), and analyzed the structural changes of TEPA that was collected from the 60th cycles with ¹³C NMR, ¹H MNR and FT-IR (Supplementary Fig. 12). The results of the SEM, TEM, and N₂ sorption analysis reveal that the spherical morphology, interior architecture, and pore structure were well maintained although the surface area had a slight decrease (from 539 to 459 m² g⁻¹). Comparing the ¹³C NMR spectra of the fresh TEPA and the recovered TEPA, one can find that the various C signals have no significant change except that a very weak signal at 163.1 ppm appears for the recovered TEPA. These results indicate that the structure and composition of TEPA were largely maintained although a small number of urea groups were formed[52]. This is further supported by the comparative ¹H NMR and FT-IR results of the fresh TEPA and the recovered TEPA. The structural maintenance is consistent with our above inference that the loss of adsorption capacity is mainly caused by the TEPA evaporation. To further evaluate the amine loss, we compared SLHSP and TEPA/MSP mixture, both of which were treated with N₂ at a high flow rate (at 100 °C). After the same treatment SLHSP had a mass loss of 8%, whereas TEPA/MSP mixture suffered from a 20% mass drop (Fig. 6e). Such a drastically

reduced amine loss should be attributed to the unique structure of SLHSP, in which the hydrophobic shell prevented the hydrophilic TEPA molecules from passing through (Fig. 6f)[49]. Encouraged by the above results, we further examined multiple cycles in the presence of moisture. The simulated flue gas was pre-humidified by first bubbling it through a water bath at room temperature (40% relative humidity was obtained). The fresh SLHSP gave a sorption capacity of 6.1 mmol g⁻¹, higher than that found in the absence of water. Consistent with the reported results, this water-induced improvement in sorption capacity is due to the formation of bicarbonates[52]. Although over the consecutive cycles the CO₂ capacity dropped slightly, the sorption capacity was still observed to be as high as 4.5 mmol g⁻¹ for the 60th cycle (Fig. 7a). The SEM images of the spent sorbent showed that the SLHSPs are robust enough to survive the 60 cycles, with their size, morphology and surface structure all being well preserved (Fig. 7b). The outstanding water-tolerant stability may be related to the hydrophobic layer on the surface of our superparticles. This water-tolerant stability was further evaluated through comparison with TEPA/MSP mixture using steam treatment (106 °C, 24 h, as illustrated in Fig. 7c). Following the steam treatment, no changes in the morphology of SLHSP were observed, and the surface still kept the hydrophobic property (Fig. 7d). The loss of sorption capacity for SLHSP was 14%, whereas the TEPA/MSP mixture had a far greater decrease of 31% (Fig. 7e). These findings further confirmed the importance of the hydrophobic silica layer on the surface.

As another proof of the concept, our strategy is adaptable to the case of polyethyleneimine (PEI, $M_W = 800$). The SLHSPs prepared from PEI possess uniform micro-spherical morphology and relatively smooth surface (Supplementary Fig. 13a and b), even in the presence of the high dosage of PEI (ca. 82 wt%, Supplementary Fig. 13d). From the cross-sectional SEM image, one can see plenty of void spaces stemming from the interstices between MSPs (Supplementary Fig. 13c). The sorption capacity of

PEI-based SLHSPs was determined to be as high as 5.7 mmol g$^{-1}$. Over the subsequent 19 adsorption–desorption cycles, the $CO_2$ sorption capacity was always maintained at ca. 5.2–5.7 mmol g$^{-1}$, without apparent loss (Supplementary Fig. 14a). After 20 cycles the amine loss was examined by TGA to be 2.3 wt% relative to the fresh PEI-based SLHSPs (Supplementary Fig. 14b), highlighting the potential for practical applications.

## Discussion

We have successfully developed a simple but effective method to fabricate a class of solid–liquid hybrid superparticles based on assembly of liquid and nanoparticles at microscale. This method was demonstrated to be controllable in size of superparticles, their interior architecture, as well as their mechanical strength. The unexpected high mechanical strength was attributed to the strong interfacial adsorption of particles and strong interplays between arrangement of nanoparticles and the interior liquid, which is also supported by theoretical analysis. Another attractive feature of the proposed superparticle is high accessibility of the interior liquid, because of the hierarchical microstructures created by judicious combination of liquid and solid. As a proof of the concept, the hybrid superparticles could be used directly in fixed bed for $CO_2$ capture demonstrating excellent sorption capacity (6.1 mmol g$^{-1}$), high sorption rates and long-term stability over multiple cycles (60 times), even in the presence of moisture. Interestingly, SLHSP could effectively retard the amine loss due to its hydrophobic shell, which is virtually unattainable for the presently reported amine-impregnated sorbents in the literature. In view of their excellent performance, the developed SLHSP stand out in simultaneously enhancing many desirable properties of micron-sized sorbents. More importantly, it was found that the sorption performance is crucially related to the microstructure, thus shedding light on to the structure–performance relationship for this material. This study opens up a horizon in design of solid–liquid hybrid materials for further innovative applications.

## Methods

**Fabrication of TEPA marble**. Typically, 0.55 g TEPA was added into 1.25 g distilled water under stirring, yielding a 30 wt% TEPA water solution. This solution was transferred into a syringe with an orifice diameter of 50 μm and then continuously injected onto a bed made of MSP-Os at 0.3 ml min$^{-1}$. After rolling on the bed, aqueous TEPA marbles were obtained.

**SLHSP fabrication**. A given amount of MSP was added to the above 30 wt% TEPA water solution under stirring, yielding a suspension. This suspension was transferred into a syringe with an orifice diameter of 50 μm and continuously injected onto a bed made of MSP-Os at 0.05, 0.15 or 0.30 ml min$^{-1}$. After rolling on the bed, MSP-containing liquid marbles were obtained. The MSP-containing liquid marbles was heated at 60 °C to evaporate the water under vacuum (10 h), eventually leading to SLHSPs.

**Theoretical analysis of the relationship between drop size and the suspension viscosity**. Mechanistic equations for the prediction of the dispersion drop size are based on the contribution of Hinze, who proposed that drop breakup resulted from the balance between the disruptive force caused by the variation of dynamic pressure on the drop surface and the resistive forces due to interfacial tension and dispersed-phase viscosity, which is a generalized Weber group, $W_e$, and viscosity group $V_i$. When both interfacial tension and dispersed-phase viscous forces contribute to drop stability, he proposed the following form of function[53]:

$$W_e = c[1 + \varphi(V_i)] \tag{8}$$

Calbrese et al. derived the following correlation for prediction of the Sauter mean diameter

$$\frac{d}{d_0} = [1 + BV_i]^{3/5} \tag{9}$$

$$V_i = \mu_d \varepsilon^{1/3} d^{1/3}/\sigma \tag{10}$$

Where $\sigma$ is the interfacial tension, $\mu_d$ is the viscosity of the dispersed phase. It indicates that the $d$ increased as dispersed-phase viscosity increased.

**Sorption kinetics**. To analyze the $CO_2$ sorption kinetics, the time-dependent adsorption profiles of SLHSP were fitted with a pseudo-first-order kinetic model, in which $k_1$ (min$^{-1}$) is the adsorption rate constant, and $Q_t$ (mmol g$^{-1}$) and $Q_e$ (mmol g$^{-1}$) are the adsorption capacities at a given time $t$ and the equilibrium time, respectively.

$$Q_t = Q_e[1 - e^{-k_1 t}] \tag{11}$$

**Cyclic adsorption–desorption measurement**. In a typical sorption measurement, the sorbent was packed in a quartz column (diameter of 0.7 cm and length of 10 cm), at bottom of which glass wool was installed. The outlet of packed bed was connected to a $CO_2$ analyzer for online determination. Before $CO_2$ adsorption, the sorbent was activated at 100 °C for 60 min in dry $N_2$ and then cooled down to 75 °C. Subsequently, the inlet gas was switched to 15% $CO_2$ with a flow rate of 25 ml min$^{-1}$. The $CO_2$ concentration of the effluent gas was monitored using the $CO_2$ analyzer. After sorption for a certain time, the inlet gas to the packed bed was switched back to $N_2$ and the packed bed was heated to 100 °C for $CO_2$ desorption. Following the same procedure, a blank control measurement was carried out using the bed packed only with glass wool to evaluate the dilution effect from gas changes. The $CO_2$ concentration profile after removal of the background was used for the sorbent capacity calculations.

## Data availability

The data that support the plots within this paper and other findings of this study are available from the corresponding authors upon reasonable request.

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

## Acknowledgements

This work is supported by the Natural Science Foundation of China (21733009, 21603128, 21733009, 21573136, and U1510105), the Key Research and Development Project of Shanxi Province of China (201603D312003), and the Foundation of State Key Laboratory of Coal Conversion (Grant No. J19-20-609). S.V. Lishchuk acknowledges support from the Erwin Schrödinger International Institute for Mathematics and Physics (ESI), Vienna.

## Author contributions

X.R. performed the experiments and characterization. R.E. and S.V.L. established the physical model. H.C., N.Z., and F.X. gave advices on the sorption experiment. H.C. and F.C. contributed to the discussion. H.Y. conceived and designed the experiments. The manuscript was written through contributions of all authors.

## Additional information

**Competing interests:** The authors declare no competing interests.

