## [Peer Review File · Nature Communications]

Reviewers' comments:

Reviewer #1 (Remarks to the Author):

In the present work, the authors synthesized a solid-liquid hybrid superparticle for CO₂ capture application. The key synthesis idea involves the assembly of hydrophobic silica nanoparticles on the liquid droplet surface, and co-assembly of hydrophilic silica nanoparticles and tetraethylenepentamine (TEPA) within the interior of the resulting droplets. In my viewpoint, the synthesis idea based on the authors' expertise on rheology is interesting. However, in terms of CO₂ capture application, the use of TEPA is highly problematic, because it can evaporate continuously at the elevated temperature of desorption condition. Large CO₂ capacity of TEPA is not surprising because it has high-density primary and secondary amines. The serious weaknesses of TEPA are the continuous evaporation due to its low molecular weight as well as difficult regeneration (desorption) due to large CO₂ heat of adsorption. In this study, the authors used N₂ purging for regeneration and thus could use moderate regeneration temperature. However, such conditions cannot be used in practical CO₂ capture processes.

I feel that the novelty of this work exists not on the CO₂ adsorption application, but on the material synthesis. However, the authors should have used at least a PEI (polyethyleneimine) for the material preparation rather than TEPA. I strongly request the authors to carry out the experiments again using PEI instead of TEPA. If the methodology can be generalized to PEI having higher viscosity than TEPA, I would reconsider the acceptance of the present manuscript. Please note that even the PEI has many problems in CO₂ capture applications (low chemical stabilities and large regeneration heat due to H₂O coadsorption). However, if the present strategy can be applied to PEI, it can be also applied to more advanced amine polymers developed recently. The authors also need to mention in the revised manuscript that the PEI is just used for proof of concept, and the strategy can be generally applied to other amine polymers with improved properties.

Reviewer #2 (Remarks to the Author):

This manuscript describes the synthesis of amine-silica hybrid particles and their use in CO₂ capture. The preparation/assembly route of these particles is interesting and rather niche. Also the material showed a high CO₂ adsorption capacity with an impressive recycling regime. There are interesting features in the manuscript but there are also some queries to be addressed prior to publication.

Major comments:

1. In the abstract, the authors highlighted the CO₂ adsorption capacity (6.1 mmol/g) as "outstanding". MOF has shown >7 mmol/g while PEI/MCM41 (a compatible material to this work) has also shown ca. 5 mmol/g. Both were reported several years ago. I am not sure "outstanding" is accurately used here. Also, the authors mentioned "excellent sorption rate". Does it refer to the CO₂ adsorption kinetics? Based on the kinetic data shown in 5E, the samples took around 20 mins to achieve near maximum. This is indeed rather slow. Nonetheless, the maximum capacity achieved here is good. I am impressed by the number of cycles that had been repeated. That takes some work.
2. Figure 1 was supposed to show how the material is formed but it is far from clear. There are arrows that had no labels and it is difficult to follow.
3. There are hydrophobic and hydrophilic compartments in the particle. But what are the function(s) of these compartments? Why does it need both compartments? How does it work stepwisely? It was not clearly explained.

4. The authors spent a large section on the mechanical stability of the material. Yes, it is important. But I think the authors should also strengthen the CO₂ adsorption section, which was highlighted in the title of the manuscript (not mechanical stability per se).
5. One key feature of CO₂ adsorption is selectivity, in particular over N₂ or CH₄. Can the authors demonstrate that?
6. Another issue is the "CO₂ adsorption efficiency", ie CO₂:N ratio. That will show how much amine is directly engaging with CO₂ molecules. There is no such information in the manuscript.
7. It is excellent to see the material lasted 60 cycles. The authors had examined the "spent" particles but only with SEM. I will be surprised if SEM can tell us a lot except morphology. Since the authors had gone that far for testing, the material should be thoroughly tested in terms of TGA (ie composition, amine content etc), FTIR etc. One interesting question is that are the TEPA component still TEPA after many cycles of heating and cooling? Has the TEPA polymerised?

Reviewer #3 (Remarks to the Author):

This is a well prepared and comprehensive report which, in my view, also represents a significant fundamental advance in the development of composite materials for adsorption purposes (and, potentially, other purposes). I believe it is of interest to Nature Communications readers and that it should be published subject to addressing the issues detailed below. Mostly these relate to improving the clarity of the discussion.

Lines 64-65:

I question the authors' assertion that the highest CO₂ capacity ever recorded by amine supported silica materials is 4.3 mmol.g⁻¹. See, for example, Qi et al, Nat. Commun. 2014, 5, 5796. The introduction should be revised accordingly.

Lines 87-88:

I am concerned that the authors should provide a little more detail about the nature of the MSP and MSP-O materials that they have utilised as components for the creation of the marbles. There is a description of the synthetic methods in the supplementary material and some basic N₂ adsorption isotherms, but they have not reported the full BJH pore size distributions from this data (only average pore size data). Is this pore size data consistent with the TEM/SEM data? I feel this warrants a little more discussion.

Line 106: 'discussed' should be 'discuss'. Remove 'the' from 'the TEPA'.

Further, there seems to be no detailed description in the Supplementary material about the specifics of the preparation of TEPA marbles. What is the concentration of TEPA in water? What was the size of the droplets? Perhaps it was the same as for the SLHSP, but this is not explicitly stated. Further clarity is required.

Line 115, also line 119: Does 'commonly used mesoporous silica' refer to the prepared MSP or some other form of mesoporous material. This needs to be made clear. If it is MSP, why isn't it referred to by this acronym? Also, it is not clear whether this 'commonly used mesoporous silica' refers to a suspension of silica in the syringe or silica on the bed (glass slide). These details must also be made very clear to the reader.

Line 144: 'somewhat' should be 'some'

Lines 148-149: I don't understand the logic of how the images in Supp Fig 5a confirm this point,

since one would need to see a direct comparison of the non-calcined and calcined particles. Supp Fig 5a should, I think, be revised to provide this comparison.

Lines 168: 'widened' should probably be 'broadened'

Lines 169-173: The reference to 'grafted TEPA' is unclear. How was this 'grafted TEPA' prepared. There is no reference quoted and I can't find anything in the supplementary material either. This requires clarification.

Line 196: 'adoption' should be 'adsorption'

Line 201: 'somewhat' should be 'some'

Lines 304-314: This section refers to SBA-15. There is no indication where this SBA-15 has come from. Did the authors make it, was it purchased, was it provided by another lab? Also how does the SBA-15 compare to the MPS that the authors have made and characterised. The synthetic procedure for MPS is reasonably similar to SBA-15 – so, in this reviewer's view, there may not be much distinction between them. The extent of the distinction or similarity, requires clarification. Also, the ratio between microporosity and mesoporosity can be varied in SBA-15. Is this also the case for MPS? Could this be important in terms of the optimal % of MPS in the MPS/TEPA mixture?

Line 320: 'larger' should be 'large'

Line 387: 'keep' should be 'kept'

Line 399: According to my understanding the authors have not trialled the SLHSP in a 'fluidized bed' as implied by this sentence. They have only showed that the SLHSP should theoretically withstand the forces encountered in a fluidized bed. This sentence/paragraph needs revision.

Supplementary Material:

8. Steam Treatment: Is DI deionized water? Please specify.

Supplementary Fig 5: The scale bar on fig 5b is seemingly identical to the scale bar on fig 5a – yet b is supposed to be a magnification of a. This requires revision.

Manuscript ID: NCOMMS-18-37890.

Manuscript Type: Article

Title: "Liquid Marble-Derived Solid-Liquid Hybrid Superparticles for CO₂ Capture"

Author(s): Xia Rong, Rammile Ettelaie, Sergey V. Lishchuk, Huaigang Cheng, Ning Zhao, Fukui Xiao, Fangqin Cheng, and Hengquan Yang

Response to Referees

Reviewer 1

1. *In the present work, the authors synthesized a solid-liquid hybrid superparticle for CO₂ capture application. The key synthesis idea involves the assembly of hydrophobic silica nanoparticles on the liquid droplet surface, and co-assembly of hydrophilic silica nanoparticles and tetraethylenepentamine (TEPA) within the interior of the resulting droplets. In my viewpoint, the synthesis idea based on the authors' expertise on rheology is interesting. However, in terms of CO₂ capture application, the use of TEPA is highly problematic, because it can evaporate continuously at the elevated temperature of desorption condition. Large CO₂ capacity of TEPA is not surprising because it has high-density primary and secondary amines. The serious weaknesses of TEPA are the continuous evaporation due to its low molecular weight as well as difficult regeneration (desorption) due to large CO₂ heat of adsorption. In this study, the authors used N₂ purging for regeneration and thus could use moderate regeneration temperature. However, such conditions cannot be used in practical CO₂ capture processes. I feel that the novelty of this work exists not on the CO₂ adsorption application, but on the material synthesis. However, the authors should have used at least a PEI (polyethyleneimine) for the material preparation rather than TEPA. I strongly request the authors to carry out the experiments again using PEI instead of TEPA. If the methodology can be generalized to PEI having higher viscosity than TEPA, I would reconsider the acceptance of the present manuscript. Please note that even the PEI has many problems in CO₂ capture applications (low chemical stabilities and large regeneration heat due to H₂O coadsorption). However, if the present strategy can be applied to PEI, it can be also applied to more advanced amine polymers developed recently. The authors also need to mention in the revised manuscript that the PEI is just used for proof of concept, and the strategy can be generally applied to other amine polymers with improved properties.*

Response: Thank you very much for your valuable comments and suggestions. We completely agree with your view of the choice of organic amines. From the viewpoint of amine loss, PEI (polyethyleneimine) is absolutely a better choice for CO₂ capture because of its lower volatility in comparison to tetraethylenepentamine (TEPA). It is worth mentioning that in this work, with one of aims at investigating the impact of the hydrophobic shell of the developed SLHSP on the amine loss during multiple adsorption-desorption cycles, we chose TEPA. Nevertheless, according to

your constructive suggestion, PEI with an average molecular weight of ($M_w = 800$) was used to examine our proposed method.

As the appearance and SEM images in Supplementary Figs. 12a and 12b show (also shown below), the SLHSPs prepared from PEI possess uniform spherical morphology and relatively smooth surface, even in the presence of the high dosage of PEI (*ca.* 82 wt%, Supplementary Fig. 12d). From the cross-sectional SEM image, one can see plenty of void spaces stemming from the interstices between MSPs (Supplementary Fig. 12c). As such, we can reach a conclusion that our method is really applicable to the highly viscous PEI as well.

Supplementary Figure 12 | Characterization of the SLHSPs prepared with PEI. (a) Appearance of SLHSPs prepared with PEI. (b) SEM images of PEI-based SLHSPs. (c) SEM images of the cross-section of a single PEI-based SLHSP particle showing the interior structure. (d) TGA profile of PEI-based SLHSPs (100 ml min^{-1} air, 10 °C min^{-1}).

With the uniform PEI-based SLHSP in hand, we measured its sorption capacity and recyclability in a fixed-bed reactor using a simulated flue gas (15 vol % CO_2 in N_2 with relative humidity 40%). The obtained results are presented in Supplementary Fig. 13a (also displayed below). The sorption capacity of PEI-based SLHSP was determined to be as high as 5.7 mmol g^{-1} . Over the subsequent 19 adsorption-desorption cycles, CO_2 sorption capacity was always maintained at *ca.* $5.2\text{--}5.7 \text{ mmol g}^{-1}$, without apparent loss. After 20 cycles the amine loss was examined by TGA to be 2.3 wt % as compared to the fresh PEI-based SLHSP (Supplementary Fig.

13a). Such an amine loss is lower than the case of TEPA, making it better to regenerate the sorbent at higher temperatures. The negligible amine loss is attributed to the low volatility, which verify your inference.

Supplementary Figure 13 | (a) Multiple cycles of CO₂ adsorption–desorption over PEI-based SLHSPs. (b) TGA profile of PEI-based SLHSPs after 20 cycles.

We have included these results with necessary discussion in our revised manuscript (page 16 to 17, highlighted in yellow). “As another proof of the concept, our strategy is adaptable to the case of polyethyleneimine (PEI, $M_w = 800$, see Supplementary Experimental Section). The SLHSPs prepared from PEI possess uniform spherical morphology and relatively smooth surface (Supplementary Figs. 12a and 12b, even in the presence of the high dosage of PEI (ca. 82 wt%, Supplementary Fig. 12d). From the cross-sectional SEM image, one can see plenty of void spaces stemming from the interstices between MSPs (Supplementary Fig. 12c). The sorption capacity of PEI-based SLHSP was determined to be as high as 5.7 mmol g^{-1} . Over the subsequent 19 adsorption-desorption cycles, the CO₂ sorption capacity was always maintained at ca. $5.2\text{--}5.7 \text{ mmol g}^{-1}$, without apparent loss (Supplementary Fig. 13a). After 20 cycles the amine loss was examined by TGA to be 2.3 wt% relative to the fresh PEI-based SLHSP (Supplementary Fig. 13b), highlighting the potential for practical applications. The negligible amine loss is attributed to the low volatility of PEI.”

Reviewer 2

We are grateful for your positive comments and valuable suggestions. We answer your questions as follows.

1. In the abstract, the authors highlighted the CO₂ adsorption capacity (6.1 mmol/g) as “outstanding”. MOF has shown $>7 \text{ mmol/g}$ while PEI/MCM41 (a compatible material to this work) has also shown ca. 5 mmol/g . Both were reported several years ago. I am not sure “outstanding” is accurately used here. Also, the authors mentioned “excellent sorption rate”. Does it refer to the CO₂ adsorption kinetics? Based on the kinetic data shown in 5E, the samples took around 20 mins to achieve near maximum. This is indeed rather slow. Nonetheless, the

maximum capacity achieved here is good. I am impressed by the number of cycles that had been repeated. That takes some work.

Response: We fully agree with you that many reported solid sorbents exhibit high adsorption capacity. However, most of these mentioned solid adsorbents are particles with nanometer sizes. The tiny particles are difficult to install directly into industrial settings (for example, fixed bed reactors) because there is a high risk to plug fixed beds with great pressure drop. To meet the practical requirements, these nanoparticles must be shaped to large sized objects with uniform morphology through compression or spray-drying with assistance of binders. However, addition of binders often leads the sorbents both to being unpredictable in structure and to decrease in sorption capacity. To solve these problems, we attempt to prepare micron-to-millimeter sized sorbents, which are amenable to being packed directly in fixed-bed reactors. Compared to the reported micron-to-millimeter sized sorbents, our adsorption capacity is impressive. It is the same case with the adsorption rate. When considering the large size of our SLHSP sorbent, the rate is still good. For example, the rate is still higher than that of the nanosized TEPA-impregnated mesoporous materials (Fig. 5E).

All the same, we have changed the word “outstanding” to “excellent” in our revised version according to your suggestion.

2. Figure 1 was supposed to show how the material is formed but it is far from clear. There are arrows that had no labels and it is difficult to follow.

Response: Thank you very much for your advice. In Fig. 1A, the curved arrows represent rolling of solid-liquid suspension droplets on the bed made of MSP-O powders and picking up the MSP-O particles by the droplet surface. The arrow in Fig. 1B means that the slow evaporation of water inside the marbles causes isotropic shrinkage of the marbles and co-assembly of MSP with TEPA within the liquid marbles. According to your suggestion, we have modified the drawing of Fig. 1 for more clarification, as shown below.

Figure 1 | Schematic illustration of the preparation of SLHSPs. (A) Dispensing MSP-containing aqueous TEPA droplets using a syringe pump and rolling the droplets on a bed made of MSP-O, SEM image of MSP (upper panel), TEM image of MSP (lower panel). (B) Water evaporation causing isotropic shrinkage of the marbles and co-assembly of MSP with TEPA, and appearance of SLHSPs (in a vial).

3. *There are hydrophobic and hydrophilic compartments in the particle. But what are the function(s) of these compartments? Why does it need both compartments? How does it work stepwisely? It was not clearly explained.*

Response: The hydrophobic MSP-Os were used to fabricate liquid marbles because the hydrophobic nanoparticles can be attached at the gas/liquid interface through interfacial adsorption, forming a hydrophobic shield layer around liquid droplets. This shield layer can prevent the droplet coalesce, being favorable to maintaining of spherical morphology. As evidenced by our experiment, evaporation of the droplet (without MSP-Os on the surface) led to a doughnut-like shape due to the so-called coffee loop effect, whereas in the presence of MSP-Os uniform microspheres were obtained. Moreover, the hydrophobic layer made of MSP-O can prevent the evaporation of the enclosed hydrophilic TEPA during the course of CO₂ desorption, as shown in Fig. 6 in the revised manuscript.

The hydrophilic MSPs can be well dispersed in hydrophilic TEPA, and are subsequently subjected to assembly within the liquid marble due to the hydrogen-bonding interactions. In doing so, it is possible to obtain hierarchical interior structures and “porous liquid”. As the SEM image of the cross-section of SLHSP shows (Supplementary Fig. 5), plenty of macropores stemming from the interstices between MSPs were observed. That is to say, without MSP it is impossible to yield void spaces within liquid droplets. At the same time, MSPs act as pillars to support the superparticle architectures, thus reinforcing the solid-liquid hybrid superparticle.

The respective roles of MSP-Os and MSP are discussed on pages 6, 7, and 16.

4. *The authors spent a large section on the mechanical stability of the material. Yes, it is important. But I think the authors should also strengthen the CO₂ adsorption section, which was highlighted in the title of the manuscript (not mechanical stability per se).*

Response: According to your advice, we strengthen discussion of the CO₂ adsorption in four aspects. Firstly, to further test our new concept, we chose PEI ($M_w = 800$) to prepare solid-liquid hybrid superparticles again, and then discussed the adsorption performances (page 16 to 17, highlighted in yellow). “As another proof of the concept, our strategy is adaptable to the case of polyethyleneimine (PEI, $M_w = 800$, see Supplementary Experimental Section). The SLHSPs prepared from PEI possess uniform spherical morphology and relatively smooth surface (Supplementary Figs. 12a and 12b, even in the presence of the high dosage of PEI (ca. 82 wt%, Supplementary Fig. 12d). From the cross-sectional SEM image, one can see plenty of void spaces

stemming from the interstices between MSPs (Supplementary Fig. 12c). The sorption capacity of PEI-based SLHSP was determined to be as high as 5.7 mmol g^{-1} . Over the subsequent 19 adsorption-desorption cycles, the CO_2 sorption capacity was always maintained at ca. $5.2\text{--}5.7 \text{ mmol g}^{-1}$, without apparent loss (Supplementary Fig. 13a). After 20 cycles the amine loss was examined by TGA to be 2.3 wt% relative to the fresh PEI-based SLHSP (Supplementary Fig. 13b), highlighting the potential for practical applications. The negligible amine loss is attributed to the low volatility of PEI.”

Secondly, the SLHSP after 60 adsorption-desorption cycles has been characterized with SEM, TEM, ^{13}C NMR, ^1H NMR and FT-IR (Supplementary Figs. 10 and 11). With these supplementary results, we discussed the morphology and structure of the “spent” SLHSP and the reasons for the decrease in adsorption capacity over 60 cycles. These contents are supplied on page 15 in our revised manuscript (highlighted in yellow). “To clarify the structural and compositional changes of SLHSP over multiple cycles, we characterized SLHSPs after 60 cycles with SEM, TEM and N_2 physical sorption analysis (Supplementary Fig. 10), and analyzed the structural changes of TEPA that was collected after 60th cycles with ^{13}C NMR, ^1H NMR and FT-IR (Supplementary Fig. 11). The results of the SEM, TEM and N_2 sorption analysis reveal that the spherical morphology, interior architecture, and pore structure were well maintained although the surface area had a slight decrease (from 539 to $459 \text{ m}^2 \text{ g}^{-1}$) after 60 cycles. Comparing the ^{13}C NMR spectra of the fresh TEPA and the recovered TEPA, one can find that the various C signals have no significant change except that a very weak signal at 163.1 ppm appears for the recovered TEPA. These results indicate that the structure and composition of TEPA were largely maintained although a small number of urea groups were formed⁵². This is further supported by the comparative ^1H NMR and FT-IR results of the fresh TEPA and the recovered TEPA. The fraction of the formed urea is negligible, which can be explained by the fact that we conducted CO_2 desorption at a moderate temperature ($100 \text{ }^\circ\text{C}$). The structural maintenance is consistent with our above inference that the loss of adsorption capacity is mainly caused by the TEPA evaporation.”

Thirdly, the amine efficiency has been discussed on page 13 in our revised manuscript (highlighted in yellow). “The amine efficiency (the CO_2/N ratio) of the SLHSP with 23 wt% MSP is estimated to be 0.25, which is lower than the theoretical amine efficiency (0.5 mol CO_2 per mol N under dry conditions). The discrepancy can be explained by the fact that the high amine loading leads to the lowered accessibility of a portion of amines. Although the amine efficiency is lower than that of the reported sorbents prepared by grafting amine, it is slightly higher than those of nanosized amine-impregnated sorbents (0.18) and our amine-impregnated MSP (0.23)⁴⁸”

Finally, we have increased the discussion on the CO_2/N_2 selectivity (page 13, highlighted in yellow). “The CO_2/N_2 selectivity is estimated to be up to 187 from the TG-MS measurement. The high selectivity is due to the strong CO_2 –amine chemical interactions.”

5. One key feature of CO₂ adsorption is selectivity, in particular over N₂ or CH₄. Can the authors demonstrate that?

Response: According to your suggestions, the CO₂/N₂ selectivity was measured with TG-MS analyzer during the course of CO₂ desorption. The discussion of the CO₂/N₂ selectivity has been added in our revised manuscript (page 13, highlighted in yellow). “*The CO₂/N₂ selectivity is estimated to be up to 187 from the TG-MS measurement. The high selectivity is due to the strong CO₂-amine chemical interactions.*”

6. Another issue is the “CO₂ adsorption efficiency”, ie CO₂:N ratio. That will show how much amine is directly engaging with CO₂ molecules. There is no such information in the manuscript.

Response: We agree with you that “amine efficiency” is important. The discussion of amine efficiency is supplied in our revised version. Amine efficiency is defined as the ratio of the moles of captured CO₂ to the total moles of N on the sorbent. Since the TEPA content in SLHSP is ~80.3 wt%, the N content was estimated to be 21.2 mmol N g⁻¹. Under dry conditions, the amine efficiency (the CO₂/N ratio) of the SLHSP is *ca.* 0.25, which is lower than the theoretical amine efficiency (0.5 mol CO₂ per mol N under dry condition). The discrepancy could be explained by the fact that the high amine loading leads to the lowered accessibility of a portion of amines.

We have increased the discussion of amine efficiency in our revised manuscript (page 13, highlighted in yellow). “*The amine efficiency (the CO₂/N ratio) of the SLHSP with 23 wt% MSP is estimated to be 0.25, which is lower than the theoretical amine efficiency (0.5 mol CO₂ per mol N under dry conditions). The discrepancy can be explained by the fact that the high amine loading leads to the lowered accessibility of a portion of amines. Although the amine efficiency is lower than that of the reported sorbents prepared by grafting amine, it is slightly higher than those of nanosized amine-impregnated sorbents (0.18) and our amine-impregnated MSP (0.23)⁴⁸*”

7. It is excellent to see the material lasted 60 cycles. The authors had examined the “spent” particles but only with SEM. I will be surprised if SEM can tell us a lot except morphology. Since the authors had gone that far for testing, the material should be thoroughly tested in terms of TGA (ie composition, amine content etc), FTIR etc. One interesting question is that are the TEPA component still TEPA after many cycles of heating and cooling? Has the TEPA polymerised?

Response: Thank you very much for your comments. In addition to the SEM images of the recovered SLHSP, TGA results have been provided in our manuscripts, which revealed a 15 wt% amine loss after 60 cycles, compared to the fresh SLHSP (Fig. 6D). The loss of sorption capacity after 60 cycles was estimated to be *ca.* 14% of the initial capacity (Fig. 6B). Based on the results of the amine loss and the adsorption capacity decrease, we can infer that the decrease in adsorption capacity is mainly caused by the TEPA evaporation during multiple cycles.

According to your suggestions, we further characterized the recovered SLHSPs after washing with methanol with SEM, TEM and N₂ physical sorption analysis (Supplementary Fig. 10, as presented below), and characterized TEPA that was collected from the SLHSPs after 60th cycles

with ^{13}C NMR, ^1H MNR and FT-IR (Supplementary Fig. 11, as presented below). The results of the SEM, TEM and N_2 physical sorption analysis revealed that the spherical morphology, interior architecture, and pore structure were well maintained although the surface area had a light decrease (from 539 to $459\text{ m}^2\text{ g}^{-1}$) after 60 cycles.

Supplementary Figure 10 | Characterization of the SLHSPs after 60 cycles (washing with methanol). (a) SEM image. (b) Magnified SEM image. (c) TEM image. (d) Magnified TEM image. (e) N_2 adsorption–desorption isotherm. (f) BJH pore size distribution.

Furthermore, comparing the ^{13}C NMR spectra of the fresh TEPA and the recovered TEPA, one can find that various C signals have no significant change except that a very weak signal at 163.1 ppm emerges for the recovered TEPA. These comparative results indicate that the structure of TEPA was largely maintained after multiple cycles although a small number of urea groups were formed. This is further supported by the comparative ^1H NMR and FT-IR results of the fresh TEPA and the recovered TEPA. The formation of urea (leading to polymerization) is consistent with the finding reported in literature (Ref. 52). The amount of the formed urea is negligible, which can be explained by the fact that we conducted CO_2 desorption at a moderate temperature ($100\text{ }^\circ\text{C}$). The structural maintenance is in accord with our above inference that the loss of adsorption capacity is mainly caused by the TEPA evaporation.

According to your advice, we have added these discussions in the revised manuscript (page 15 to 16, highlighted in yellow). “To clarify the structural and compositional changes of SLHSPs over multiple cycles, we characterized SLHSPs after 60 cycles with SEM, TEM and N_2 physical sorption analysis (Supplementary Fig. 10), and analyzed the structural change of TEPA that was collected from the 60th cycles with ^{13}C NMR, ^1H MNR and FT-IR (Supplementary Fig. 11). The results of the SEM, TEM and N_2 sorption analysis reveal that the spherical morphology, interior architecture, and pore structure were well maintained although the surface area had a slight

decrease (from 539 to 459 $\text{m}^2 \text{g}^{-1}$) after 60 cycles. Comparing the ^{13}C NMR spectra of the fresh TEPA and the recovered TEPA, one can find that various C signals have no significant change except that a very weak signal at 163.1 ppm appears for the recovered TEPA. These results indicate that the structure of TEPA was largely maintained although a small number of urea groups were formed⁵². This is further supported by the comparative ^1H NMR and FT-IR results of the fresh TEPA and the recovered TEPA. The fraction of the formed urea is negligible, which can be explained by the fact that we conducted CO_2 desorption at moderate temperature (100 °C). The structural maintenance is consistent with our above inference that the loss of adsorption capacity is mainly caused by the TEPA evaporation.”

Supplementary Figure 11 | Characterization of the SLHSP after 60 cycles. (a) ^{13}C NMR spectra of the pristine TEPA and the TEPA collected from the SLHSPs after 60 cycles. (b) ^1H NMR spectra of the pristine TEPA and the TEPA collected from the SLHSPs after 60 cycles. (c) FT-IR spectra of the fresh SLHSPs and the SLHSPs after 60 cycles.

Reviewer 3

Thank you so much for your positive comments and suggestions. We have revised our manuscript according to your suggestions.

1. Lines 64-65: *I question the authors assertion that the highest CO₂ capacity ever recorded by amine supported silica materials is 4.3 mmol.g⁻¹. See, for example, Qi et al, Nat. Commun. 2014, 5, 5796. The introduction should be revised accordingly.*

Response: We agree with you that many developed solid sorbents have shown high adsorption capacity. Most of these mentioned solid adsorbents are particles with nanometer sizes. The tiny particles are difficult to install directly into industrial settings (for example fixed bed) because there is a high risk to plug fixed beds (great pressure drop). To meet these technical requirements, the nanoparticles must be shaped into large sized objects with uniform morphology through compression or spray-drying with assistance of binders. However, addition of binders often leads the sorbents both to being unpredictable in structure and to decrease in sorption capacity. To solve these problems, we attempt to prepare micron-to-millimeter sized sorbents, which are amenable to being packed directly in fixed-bed reactors. Compared to the reported micron-to-millimeter sized sorbents, our adsorption capacity is impressive (the value of 4.3 mmol.g⁻¹ is only for micron-to-millimeter sized sorbents). All the same, we have changed the word “*outstanding*” to “*excellent*” in the abstract in our revised version according to your suggestion.

2. Lines 87-88: *I am concerned that the authors should provide a little more detail about the nature of the MSP and MSP-O materials that they have utilised as components for the creation of the marbles. There is a description of the synthetic methods in the supplementary material and some basic N₂ adsorption isotherms, but they have not reported the full BJH pore size distributions from this data (only ave pore size data). Is this pore size data consistent with the TEM/SEM data? I feel this warrants a little more discussion.*

Response: According to your suggestion, the isotherms and BJH pore size distribution of MSP and MSP-O are included in the inset of Supplementary Fig. in 1e and f, and showed below as well. After modification with octyltrimethoxysilane, while the isotherm of MSP-O is still similar to the initial MSP indicating that the original mesoporous structure was maintained, the BJH pore size is observed to decrease from 15.5 nm to 12.7 nm. This is a result of the introduction of octyl groups into the mesoporous channels. The determined pore size of MSP is broadly consistent with the TEM observation (15–19 nm, Supplementary Fig. 1e). We have updated the corresponding discussion in our revised version (page 5, highlighted in yellow). “*MSPs have particle sizes from hundreds of nanometer to one micron, and their pore size is centered around 15–19 nm (Fig. 1A), which is broadly consistent with the BJH pore size (Supplementary Fig. 1).*”

Supplementary Figure 1 | Characterization of MSP and MSP-O. (e) N₂ adsorption–desorption isotherm and BJH pore size distribution of MSP (inset). (f) N₂ adsorption–desorption isotherm and BJH pore size distribution of MSP-O (inset).

3. Line 106: ‘discussed’ should be ‘discuss’. Remove ‘the’ from ‘the TEPA’. Further, there seems to be no detailed description in the Supplementary material about the specifics of the preparation of TEPA marbles. What is the concentration of TEPA in water? What was the size of the droplets? Perhaps it was the same as for the SLHSP, but this is not explicitly stated. Further clarity is required.

Response: Thank you for your suggestions. We have corrected these misspellings in our revised version. The procedure for preparation of the TEPA marbles has been added in the revised Supplementary Information (page S19, highlighted in yellow): “Typically, an aqueous solution of 30 wt% TEPA was transferred into a syringe with an orifice diameter of 50 μm and then continuously dropped onto a bed made of MSP-Os with the assistance of the syringe pump followed by rolling of the droplets on the bed, yielding TEPA marbles.”

The size of TEPA marbles is estimated to be *ca.* 100-200 μm from the optical micrographs (Supplementary Figs. 2b and 2c), which is smaller than that of SLHSPs (containing 9 wt% MSP). This size change could be explained in terms of the increase in viscosity when solid particles are added. The increase in viscosity provides a larger resistance for breakage, thus leading to larger sized droplets.

4. Line 115, also line 119: Does ‘commonly used mesoporous silica’ refer to the prepared MSP or some other form of mesoporous material. This needs to be made clear. If it is MSP, why isn’t it referred to by this acronym? Also, it is not clear whether this ‘commonly used mesoporous silica’ refers to a suspension of silica in the syringe or silica on the bed (glass slide). These details must also be made very clear to the reader.

Response: Thank you for your advice. Here, mesoporous silica nanospheres (MSNs) with pore size of 8 nm were used in a control experiment for explaining the reason why we chose MSP with larger pores to prepare liquid marbles. We compared the stability of liquid marble stabilized by

hydrophobicated MSPs and MSNs (modified with the same amount hydrophobic octyltrimethoxysilane). Such a comparison highlights the unique role of the chosen MSP in the formation of high-quality liquid marble, as discussed on page 6 (the first paragraph). The procedure of the MSNs preparation is included in the Supplementary Sectional Section (page S18). **“Synthesis of mesoporous silica nanospheres (MSNs).** Mesoporous silica nanospheres were synthesized through a sol-gel process in the presence of CTAC (hexadecyl trimethyl ammonium chloride) as template. 12 g of CTAC was dissolved into a mixture of 112 ml H₂O and 0.36 g triethanolamine at 60 °C. After stirring for 60 min, another mixture of 32 ml cyclohexane and 8 ml TEOS was added into the above solution. After stirring for 12 h. the resultant solid was collected by filtration, and then dried in air and calcined through the same procedure as MSP, eventually yielding MSNs.”

The detailed characterization of MSNs is presented in Supplementary Fig. 3. To make it clear, the corresponding sentence on page 5 was revised as *“In contrast, for the aqueous TEPA marbles prepared with another mesoporous silica MSNs (with pore size of 8 nm, modified with the same amount of octyl organosilane, Supplementary Fig. 3a), their surfaces were observed to be partially bare even as early as 7 min following their preparation (Supplementary Fig. 3b).”*

Supplementary Figure 3 | (a1) TEM image of MSNs. (a2) SEM image of MSNs. (a3) N₂ adsorption–desorption isotherm of MSNs. (a4) BJH pore size distribution of MSNs. (a5) Water contact angle of octyl-modified MSNs.

5. Line 144: 'somewhat' should be 'some'

Response: Thank you for careful checking. The word "somewhat" has been corrected as "some" in the revised manuscript.

6. Lines 148-149: I don't understand the logic of how the images in Supp Fig 5a confirm this point, since one would need to see a direct comparison of the non-calcined and calcined particles. Supp Fig 5a should, I think, be revised to provide this comparison.

Response: To clearly inspect a hierarchical mesoporous-macroporous network formed inside the interior of SLHSP, we removed the liquid TEPA through calcination (the calcination procedure can not affect the porous structures). As you know, the liquid TEPA is subjected to evaporation under high vacuum, which is detrimental to TEM equipment. Another consideration is that the location of the liquid in the pore will reduce the contrast between pore wall and pore during the TEM observation.

7. Lines 168: 'widened' should probably be 'broadened'

Response: Thank you. The word "widened" was replaced with "broadened" in the revised manuscript.

8. Lines 169-173: The reference to 'grafted TEPA' is unclear. How was this 'grafted TEPA' prepared. There is no reference quoted and I can't find anything in the supplementary material either. This requires clarification.

Response: According to your suggestion, the procedure for grafting TEPA onto MSP has been described in the revised Supplementary Information on page S19. "3 mmol (3-bromopropyl)trimethoxysilane was added dropwise to a 10 ml dry toluene solution of TEPA (3.6 mmol) under vigorous stirring. After refluxing for 24 h under a N₂ atmosphere, the resultant solution was added to a 50 ml suspension of MSP (1 g). The resultant mixture was further refluxed for another 24 h under a N₂ atmosphere. Afterwards, the resultant solid was filtered, washed with toluene and ethanol, and dried at 60 °C for 12 h, yielding TEPA-grafted MSP."

9. Line 196: 'adoption' should be 'adsorption'

Response: We have corrected "adoption" as "adsorption" in the revised version. Thank you!

10. Line 201: 'somewhat' should be 'some'

Response: Thank you. The word "somewhat" was corrected to "some" in the revised manuscript.

11. Lines 304-314: This section refers to SBA-15. There is no indication where this SBA-15 has come from. Did the authors make it, was it purchased, was it provided by another lab? Also how does the SBA-15 compare to the MPS that the authors have made and characterised. The synthetic procedure for MPS is reasonably similar to SBA-15 – so, in this reviewer's view, there may not be much distinction between them. The extent of the distinction or similarity, requires clarification.

Also, the ratio between microporosity and mesoporosity can be varied in SBA-15. Is this also the case for MPS? Could this be important in terms of the optimal % of MPS in the MPS/TEPA mixture?

Response: The mentioned SBA-15 was synthesized according to the previous literature [J. Am. Chem. Soc. 1998, 120, 6024–6036], which is cited in Supplementary Information. The characterization results of SBA-15 with SEM, TEM and N₂ sorption are provided in Supplementary Fig. 7. There are differences in terms of pore size, pore structure and surface area between SBA-15 and MSP, which can be found from the characterization results. The porosity of the two materials is dominantly contributed by the mesoporosity based on the *t*-Plot analysis, which is stated in the footnotes of Supplementary Fig. 7 and Supplementary Fig. 1, respectively. The aim of using SBA-15 here is to further confirm that our liquid marble method as a general approach can improve the CO₂ adsorption performance, rather than investigation of the influence of the porous structure on adsorption performance.

12. Line 320: *'larger' should be 'large'*

Response: Thank you. We have corrected “larger” as “large” in our revised manuscript.

13. Line 387: *'keep' should be 'kept'*

Response: Thank you. We have corrected “keep” as “kept” in our revised manuscript.

14. Line 399: *According to my understanding the authors have not trialled the SLHSP in a 'fluidized bed' as implied by this sentence. They have only showed that the SLHSP should theoretically withstand the forces encountered in a fluidized bed. This sentence/paragraph needs revision.*

Response: Yes, to examine the mechanical stability of SLHSP, we used a fluidized bed with a flow of N₂. In fact, in the experiments of CO₂ capture, we only used a fixed-bed reactor. According to your suggestion, we have deleted the previous word “fluidized-” in the corresponding sentence.

15. 8. *Steam Treatment: Is DI deionized water? Please specify.*

Response: Yes, DI is deionized water. The previous “DP” was replaced with “deionized water” in Supplementary Information.

16. *Supplementary Fig 5: The scale bar on fig 5b is seemingly identical to the scale bar on fig 5a – yet b is supposed to be a magnification of a. This requires revision.*

Response: Thank you for your scrutiny. The scale bar in Supplementary Fig. 5b has been corrected in the revised version.

Supplementary Figure 5 | Characterization of the calcined SLHSP. (a) SEM image of the cross section of the calcined SLHSP. **(b)** Magnified SEM image of the cross section of the calcined SLHSP. **(c)** TEM image of the calcined SLHSP. **(d)** Magnified TEM image of the calcined SLHSP.

REVIEWERS' COMMENTS:

Reviewer #1 (Remarks to the Author):

I believe that the authors carefully addressed the issues raised by the reviewers, and thus the manuscript is suitable for publication.

Reviewer #2 (Remarks to the Author):

The authors had made significant revision, taking comments from all reviewers on board. I am satisfied with the actions taken by the authors.

Reviewer #3 (Remarks to the Author):

I commend the authors for a thorough job responding to most of the reviewers' comments. There is one detail about which, in my view, still needs clarification and revision. This concerns the method of determining selectivity (p 13, 2nd last paragraph). The authors indicate that they have applied a TG-MS method to estimate the CO₂/N₂ selectivity, but they provide no details of the actual experimental protocol or the calculation procedure. The number they provide is meaningless without this context. So, this needs to be fixed prior to publication.

Manuscript ID: NCOMMS-18-37890B

Manuscript Type: Article

Title: "Liquid Marble-Derived Solid-Liquid Hybrid Superparticles for CO₂ Capture"

Author(s): Xia Rong, Rammile Ettelaie, Sergey V. Lishchuk, Huaigang Cheng, Ning Zhao, Fukui Xiao, Fangqin Cheng, and Hengquan Yang

Response to Referees

Reviewer 3

Comments: I commend the authors for a thorough job responding to most of the reviewers' comments. There is one detail about which, in my view, still needs clarification and revision. This concerns the method of determining selectivity (p13, 2nd last paragraph). The authors indicate that they have applied a TG-MS method to estimate the CO₂/N₂ selectivity, but they provide no details of the actual experimental protocol or the calculation procedure. The number they provide is meaningless without this context. So, this needs to be fixed prior to publication.

Response: Thank you very much for your positive comments and valuable suggestions. According to your suggestion, the details of the experimental procedure to determine the CO₂/N₂ selectivity have been added in the revised Supplementary Information on page S21. "The CO₂/N₂ selectivity was measured with TG-MS analyzer during the course of CO₂ desorption. Typically, about 20 mg SLHSP after completing CO₂ sorption was placed in a thermal gravimetric analyzer. After sufficiently purging the TG-MS analyzer with argon at room temperature, the sample was heated to 100 °C for CO₂ desorption. The CO₂ and N₂ released from the SLHSP were monitored with the online MS analyzer using MID mode (Multiple Ion Detection)."

The results of TG-MS measurement and calculation method have been added on page S9, as shown below.

Supplementary Figure 8 | CO₂/N₂ selectivity measurement using TG-MS method. (a) Mass spectrometry spectra of CO₂ and N₂ released from the SLHSP. **(b)** Mass spectrometry spectra of m/z 44 and m/z 28 for pure CO₂ and the corresponding peak area ratio of m/z 28 to m/z 44. The peak of m/z 44 was assigned to CO₂, and the peak area of m/z 28 was attributed to N₂ and fragment of CO₂ (CO⁺, Supplementary Figure 8a)¹. In order to determine the percentage of CO₂

fragment at m/z 28, pure CO₂ was also monitored triply by MS. As shown in Supplementary Figure 8b, the average peak area ratio of m/z 28 to m/z 44 is estimated to be 10.9%, which is very close to the value reported in standard spectrum library (11.4%). Thus, the CO₂/N₂ selectivity over SLHSP could be calculated according to the equation:

$$\text{Sel}_{\text{CO}_2/\text{N}_2} = \frac{A_{44}}{A_{28} - A_{44} \times 10.9\%}$$

where A₄₄ and A₂₈ are the peak areas of m/z 44 and m/z 28, respectively. For example, as shown in Supplementary Figure 8a, A₄₄= 3.98*10⁻⁶ and A₂₈=4.56*10⁻⁷, taking these values to the above equation, we can work out the CO₂/N₂ selectivity (187).

1. Ren, X. M., Li, H., Chen, J., Wei, L. J., Modak, A., Yang, H. Q. & Yang, Q. H. N-doped porous carbons with exceptionally high CO₂ selectivity for CO₂ capture. *Carbon* **114**, 473–481 (2017).